# Role of Lateral Hypothalamus in Acupuncture Inhibition of Cocaine Psychomotor Activity

**DOI:** 10.3390/ijms22115994

**Published:** 2021-06-01

**Authors:** DanBi Ahn, Han Byeol Jang, Suchan Chang, Hyung Kyu Kim, Yeonhee Ryu, Bong Hyo Lee, Sang Chan Kim, Kyle B. Bills, Scott C. Steffensen, Yu Fan, Hee Young Kim

**Affiliations:** 1Department of Physiology, College of Korean Medicine, Daegu Haany University, Daegu 42158, Korea; an951121@dhu.ac.kr (D.A.); star961018@naver.com (H.B.J.); c64111915@gmail.com (S.C.); badawanabi@gmail.com (H.K.K.); dlqhdgy@dhu.ac.kr (B.H.L.); 2Korean Medicine Fundamental Research Division, Korea Institute of Oriental Medicine, Daejeon 34054, Korea; yhryu@kiom.re.kr; 3Medical Research Center, College of Korean Medicine, Daegu Haany University, Gyeongsan 38610, Korea; sckim@dhu.ac.kr; 4Department of Biomedical Sciences, Noorda College of Osteopathic Medicine, Provo, UT 84606, USA; drkbills@gmail.com; 5Department of Psychology and Neuroscience, Brigham Young University, Provo, UT 84602, USA; scott_steffensen@byu.edu; 6Department of Human Anatomy and Histoembryology, Nanjing University of Chinese Medicine, Nanjing 210023, China; fanyu0309@hotmail.com

**Keywords:** lateral hypothalamus, cocaine, acupuncture, spinohypothalamic neurons

## Abstract

Acupuncture modulates the mesolimbic dopamine (DA) system; an area implicated in drug abuse. However, the mechanism by which peripheral sensory afferents, during acupuncture stimulation, modulate this system needs further investigation. The lateral hypothalamus (LH) has been implicated in reward processing and addictive behaviors. To investigate the role of the LH in mediating acupuncture effects, we evaluated the role of LH and spinohypothalamic neurons on cocaine-induced psychomotor activity and NAc DA release. Systemic injection of cocaine increased locomotor activity and 50 kHz ultrasonic vocalizations (USVs), which were attenuated by mechanical stimulation of needles inserted into HT7 but neither ST36 nor LI5. The acupuncture effects were blocked by chemical lesions of the LH or mimicked by activation of LH neurons. Single-unit extracellular recordings showed excitation of LH and spinohypothalamic neurons following acupuncture. Our results suggest that acupuncture recruits the LH to suppress the mesolimbic DA system and psychomotor responses following cocaine injection.

## 1. Introduction

Over the past 40 years, there has been a growing interest in acupuncture treatment of substance abuse around the world including cocaine addiction [1]. In 1996, the World Health Organization (WHO) recommended acupuncture as an effective treatment for substance abuse [2]. Currently, over 700 addiction recovery centers in the USA use acupuncture as an adjunctive procedure [1]. We and others have shown that acupuncture reduces drug-induced addictive behaviors by acting on the mesolimbic dopamine (DA) system [3,4,5]. We have shown that acupuncture at HT7 modulates GABAergic neurons in ventral tegmental area (VTA) and suppresses DA release in the nucleus accumbens (NAc), leading to suppression of the reinforcing properties of drugs of abuse [4,6]. Our recent study revealed that acupuncture stimulates opioidergic neurons in the arcuate nucleus of hypothalamus to attenuate substance abuse [7]. We have also proposed a peripheral mechanism that HT7 acupuncture-induced inhibition of drug-seeking behaviors is mediated via activation of specific fiber types in the ulnar nerve [8]. While the previous studies provide evidence that acupuncture modulates the mesolimbic DA system implicated in drug abuse, how the peripheral sensory afferents, during acupuncture stimulation, modulate this system needs further investigation.

The lateral hypothalamus (LH) is located, partly, in the midbrain and plays an important role in maintaining physiological and behavioral homeostasis [9]. The LH has been implicated in reward processing and addictive behaviors [9]. For example, LH neurons are activated by stimuli associated with drugs of abuse [10,11], and chemical activation of the LH reinstates an extinguished drug-seeking behavior [12]. The LH densely innervates the lateral habenula (LHb), which are mostly excitatory. Stimulation of the LH-LHb pathway inhibits the mesolimbic DA system and reward-related behaviors [13,14]. Furthermore, it is also known that the LH is involved in nociceptive processing. Nociceptive information reaches the LH through the spinohypothalamic (SHT) pathway and causes autonomic, neuroendocrine, and affective responses [15,16]. Given that the LH receives direct peripheral nociceptive inputs and is linked with the mesolimbic DA system, peripheral nociceptive signals during acupuncture stimulation may inhibit addictive behaviors through possible mediation of the LH.

To demonstrate this, the present study investigated, for the first time, whether: (1) LH mediates acupuncture effects on cocaine-induced psychomotor responses and DA release in NAc; (2) acupuncture activates spinohypothalamic neurons in cervical spinal cord; (3) activation of the LH mimics the acupuncture effects on cocaine-induced locomotion.

## 2. Results

### 2.1. The LH Mediates the Inhibitory Effects of Acupuncture at HT7 on Cocaine-Induced Locomotor Activity

Systemic injection of cocaine (15 mg/kg) increased locomotor activity, which lasted up to approximately 30 min, compared to the values before cocaine injection. When needles, inserted into acupoints HT7, LI5, or ST36, were stimulated mechanically (Figure 1A,B), acupuncture at HT7, but not at LI5 nor ST36, reduced cocaine-enhanced locomotor activity (two-way ANOVA: group factor *F* = 10.593, *p* < 0.001; time factor *F* = 38.512, *p* < 0.001; interaction *F* = 2.059, *p* = 0.004; Figure 1C,D). Reduction in cocaine locomotion by HT7 acupuncture was almost completely prevented in the rats with the LH lesions produced by bilateral injections of ibotenic acid 7 days prior (Figure 1E,F). The data were collected only in the rats with correctly placed lesions on post-mortem examination, as shown in the representative picture stained with toluidine blue (Figure 1G). These results suggest that HT7 acupuncture-induced inhibition of cocaine locomotion is mediated by a circuit containing the LH.

### 2.2. The LH Mediates the Inhibitory Effects of Acupuncture at HT7 on Cocaine-Induced 50-kHz USVs

To determine whether acupuncture at HT7 might attenuate cocaine-induced positive affective states, we explored the effect of acupuncture at HT7 on the 50 kHz USVs following cocaine injection. When cocaine (15 mg/kg) was intraperitoneally injected, the rats emitted short and high-frequency USVs of about 50 kHz (Figure 2A,B). Enhancement of 50 kHz USVs was suppressed by acupuncture at HT7 (two-way ANOVA: group factor F = 10.593, *p* < 0.001; interaction *F* = 2.059, *p* = 0.004; Figure 2C). On the other hand, in the rats with ibotenic acid bilateral lesions of the LH, acupuncture at HT7 failed to reduce cocaine-induced 50 kHz USVs (Figure 2D).

### 2.3. The LH Mediates HT7 Inhibition of Cocaine-Induced DA Release in NAc

To investigate whether acupuncture suppresses cocaine-induced DA release in NAc via the LH, in vivo FSCV was carried out in normal or LH-lesioned rats. Systemic injection of cocaine increased the peak amplitude, of electrically stimulated DA release in NAc, by about an average of 156.7 ± 6.7% over baseline at 10 min after cocaine injection. The DA release in NAc remained elevated over 30 min after cocaine injection. Acupuncture at HT7 attenuated the cocaine-induced elevation of DA release in NAc and induced a sustained decrease in basal DA release, by about 130.5 ± 9.5%, during the 32 min after cocaine injection (two-way ANOVA: group factor *F* = 0.0211, *p* = 0.885; time factor *F* = 12.320, *p* < 0.001; interaction *F* = 0.205, *p* = 1.000; Figure 3A–D). Following bilateral ibotenic acid lesions of the LH (Figure 3E), HT7 acupuncture-induced inhibition of NAc DA release was blocked (Figure 3F–H).

### 2.4. Acupuncture at HT7 Activates the LH and SHT Neurons

To determine whether acupuncture activates LH neurons or spinohypothalamic tract (SHT) fibers, in vivo extracellular recordings were performed in LH neurons and SHT axons. Figure 4 shows histograms representing extracellular recordings of LH and SHT neurons. Single-unit discharges from LH neurons were evoked during HT7 acupuncture up to about 140% over baseline (one-way repeated ANOVA: *F* = 49.513, *p* = 0.001, *n* = 20, Figure 4A–C) and returned to baseline level after termination of acupuncture. To further explore activation of SHT neurons during acupuncture, WDR neurons were identified based on a method described previously [17]. Among the WDR neurons (*n* = 64), the SHT neurons (*n* = 10), that responded to electrical stimulation (ES) of the LH, were further isolated. The baseline activities of SHT neurons were 0.26 ± 0.10 spikes/s. The activities during HT7 acupuncture increased to 4.46 ± 0.38 spikes/s (*t*-test: *t* = −10.104, *p* < 0.001, *n* = 10; Figure 4D–F). These results indicate activation of LH and SHT by acupuncture at HT7.

### 2.5. Activation of the LH Suppresses Cocaine-Induced Locomotor Activity

To evaluate whether LH activation can reduce cocaine-enhanced locomotion, we activated LH area by injecting PEPA into LH and measured locomotor activity following cocaine administration. Systemic injection of cocaine increased locomotor activity, compared to the values before cocaine injection, which was inhibited by microinjection of PEPA into the LH (PEPA; two-way ANOVA: group factor *F* = 35.782, *p* < 0.001; time factor *F* = 17.878, *p* < 0.001; interaction *F* = 2.406, *p* < 0.023; Figure 5A,B). In another set of animals, c-Fos immunostaining was performed to confirm the neuronal activity of the LH by PEPA. The PEPA group displayed significant increases of c-Fos-labelled neurons in the LH, compared to the normal or cocaine group (one-way repeated ANOVA: F(2, 10) = 10.499; Figure 5C,D).

## 3. Discussion

In the present study, acupuncture at HT7 reduced cocaine-induced psychomotor behaviors (locomotor activity and USVs) and NAc DA release, which was ablated by chemical lesion of the LH. The acupuncture effect was mimicked by chemical activation of LH by PEPA. Acupuncture excited both LH and SHT neurons. Our findings suggest that acupuncture attenuates cocaine psychomotor responses by activating LH through spinohypothalamic pathways.

In the present study, inhibitory effects of acupuncture, on cocaine-induced psychomotor responses and NAc DA release, were blocked by chemical lesion of the LH or mimicked by chemical activation of the LH by PEPA. In addition, mechanical stimulation of needles, inserted into HT7, excited LH neurons. Our present data suggest that HT7 acupuncture effects are due to activation of LH neurons. The LH has been implicated in feeding, reward, and drug seeking behaviors [12,13,18,19]. Nuclei, including those from the LH, LHb, and mesolimbic DA system, are not only reciprocally connected, but also respond to aversive events contributing to the inhibition of the reward system [14]. Glutamatergic projections from the LH innervate LHb neurons that, in turn, directly innervate midbrain rostromedial tegmental nucleus (RMTg) GABA neurons, and indirectly inhibit DA neurons [13,15]. Activation of LH-LHb glutamatergic fibers produces aversive behavior [14] or an inhibition of reward-related behavior [13,15]. Thus, it is possible that acupuncture could excite the LH neurons projecting to LHb, which would, in turn, activate RMTg GABA neurons and inhibit VTA DA neurons, leading to the suppression of cocaine-induced psychomotor behaviors. Furthermore, our in vivo extracellular recordings revealed that spinal WDR neurons, that responded to antidromical stimulation of LH, were excited by acupuncture at HT7, suggesting the activation of the spinohypothalamic tract by acupuncture. SHT neurons carry innocuous or nociceptive information to the hypothalamus that is involved in pain-related endocrine adjustments or emotional reactions [16]. Taken together, these data suggest that peripheral afferent signals from acupuncture at HT7 would be delivered to the hypothalamus through spinohypothalamic pathways and activate GABA neurons in VTA RMTg to inhibit DA neurons, thereby resulting in inhibitory effects of acupuncture on cocaine-induced psychomotor responses.

The present study reveals that acupuncture at HT7, but not ST36 nor LI5, suppressed cocaine-induced locomotor activity. This is consistent with previous studies demonstrating that acupuncture at HT7, but not LI5, reduces cocaine-induced locomotor activity [8,20] and cocaine-seeking behavior [4]. Other studies reported that acupuncture at ST36 attenuates nicotine-induced locomotor activity [21], and acupuncture at SI5 attenuates morphine-seeking behavior [22]. Thus, it may suggest that acupuncture attenuates addictive behaviors in a drug-specific manner or a point-specific manner. Our previous studies have suggested peripheral and central mechanisms underlying inhibitory actions of acupuncture at HT7 on cocaine-induced locomotor effects. Acupuncture at HT7 activates peripheral sensory afferent neurons, and the signals are transmitted via the ulnar nerve trunk [8]. Acupuncture at HT7 alleviates cocaine reduction of GABA release, and GABA neuron firing rates in the VTA, and reduces acute cocaine-induced DA release in the NAc, thereby inhibiting cocaine-induced reinforcing effects [4]. Moreover, the present study showed that acupuncture at HT7 suppressed both 50 kHz USVs and NAc DA release, following cocaine injection. USVs are classified into two distinct groups based on their sound frequency: 22 kHz and 50 kHz USVs. It is known that 22-kHz calls reflect negative affective states such as social defeat, drug withdrawal, or aversive events, while 50-kHz calls occur during a positive affective state such as play behavior, mating, addictive drugs, and rewarding events. In particular, 50 kHz is closely associated with DA release in NAc [23]. Together, these findings suggest that acupuncture can attenuate cocaine-enhanced psychomotor activity and reduce NAc DA release.

In the present study, acupuncture produced a long-lasting decrease in cocaine-induced locomotion while it activated hypothalamic neurons transiently. Our previous study showed that HT7 acupuncture activates hypothalamic neurons transiently during stimulation but increases β-endorphin levels up to 1 h after stimulation in NAc, thereby activating μ-opioid receptors, on accumbal GABA neurons, to alleviate addictive behaviors [7]. We also revealed that HT7 acupuncture increases GABA release in VTA over 40 min after stimulation and induces significant decreases of DA release in NAc over 60 min after stimulation, leading to the reduction of cocaine-enhanced locomotor activity [4]. Further, we have demonstrated that mechanical activation of this neuronal pathway, though the dorsal column-medial lemniscus pathway, elicits the same responses. Additionally, these effects can be blocked by inactivation of nicotinic acetylcholine and delta-opioid receptors in the NAc, suggesting a role for cholinergic interneurons in the NAc [24]. Thus, we suggest that a long-lasting decrease in cocaine locomotion by HT7 acupuncture would be due to modulation of neurotransmitters such as β-endorphin, GABA, acetylcholine, and DA in VTA and NAc.

In conclusion, our findings suggest that acupuncture-induced alterations, to inhibit mesolimbic DA system and cocaine enhanced psychomotor responses, are mediated through the LH by way of the spinohypothalamic tract. These findings demonstrate a need to further investigate the use of HT7 acupuncture as a promising adjunctive treatment for cocaine-use disorder.

## 4. Materials and Methods

### 4.1. Animals

Male Sprague-Dawley rats (Daehan Animal, Seoul, Korea) weighing 250–350 g were used. Home cages were kept under 12 h light-dark cycles (light off from 6:30 a.m. to 6:30 p.m.), the temperature of 22 ± 2 °C, and humidity of 40–60% with free access to food and water. All procedures were carried out in accordance with the National Institutes of Health Guide for the Care and Use of Laboratory Animals and approved by the Institutional Animal Care and Use Committee (IACUC) at the Daegu Haany University (# DHU2019-085, 22 November 2019).

### 4.2. Chemicals

Cocaine (15 mg/kg in saline; intraperitoneal [i.p.]; Macfarlan Smith Ltd., Edinburgh, UK), ibotenic acid (5 mg/mL in saline; 0.5 μL/loci; Sigma-Aldrich, MO, USA) and 2-[2,6-Difluoro-4-[[2-[(phenylsulfonyl)amino]ethyl]thio]phenoxy]acetamide (PEPA; an AMPA receptor agonist; 3 ng/side; Sigma-Aldrich, St. Louis, MO, USA) were used.

### 4.3. Acupuncture Treatment

Acupuncture was performed by using a mechanical acupuncture instrument (MAI) which consisted of a control unit and a mechanical displacer, as described in our previous publication [7]. Briefly, acupuncture treatment was performed once immediately after cocaine. While the assistant gently restrained the rat, acupuncture needles (0.10 mm in diameter, 10 mm in the length of the needle; Dongbang Medical Co., Seoul, Korea) were inserted 3 mm deep into bilateral acupoints HT7, LI5, or ST36 and stimulated for 20 s at an intensity of 1.3 m/s^2^ with the MAI. The acupoints HT7, LI5, or ST36 were positioned based on a transpositional method, which located animal acupoints on the surface of animal skin, corresponding to the anatomic sites of human acupoints [25]. HT7 is located on the transverse crease of the wrist of the forepaw, radial to the tendon of the flexor carpi ulnaris muscle. LI5 was tested, as the corresponding control point to HT7, on the opposite side of the forelimb, about 5 mm apart from HT7. ST36 is located in the anterior tibial muscle, 10 mm distal to the knee joint. Rats in the control group were handled in the same manner without needle insertion.

### 4.4. Pharmacological Lesion of Lateral Hypothalamus (LH)

As performed in our laboratory [26], ibotenic acid (0.5 μL/loci) was injected 7 days prior to the experiments. Under pentobarbital anesthesia (50 mg/kg, intraperitoneal, i.p.), two holes were drilled in the skull to access the LH (stereotaxic coordinates: AP, −2.40 mm; mL, 2.0 mm; DV, 8.8 mm) [5]. Ibotenic acid or artificial cerebro-spinal fluid (aCSF) was infused into the LH at a rate of 0.1 μL/min using a 26-gauge Hamilton syringe (Reno, NV, USA) connected to a microinjection pump (Pump 22, Harvard Apparatus, Holliston, MA, USA). The syringe was left in place, for at least 5 min, to facilitate diffusion after injection.

### 4.5. Cocaine-Induced Locomotor Activity

Locomotor activity was measured by an image analysis system (Ethovision 3.1, Noldus Information Technology, Wageningen, The Netherlands), as previously described [26,27]. Briefly, each animal was placed into a square open field box (40 cm × 40 cm × 45 cm) and monitored with an overhead video camera and video tracking software. After recording baseline activity for 30 min, animals were given an intraperitoneal injection of cocaine (15 mg/kg) and monitored up to 60 min after injection. The distance travelled during each 10-min period was analyzed. Data were expressed as a percentage of baseline activity.

### 4.6. Measurement of Ultrasonic Vocalizations

Ultrasonic vocalizations (USVs) were recorded using customized, sound-attenuating chambers as previously described [28]. The chamber consisted of two boxes to minimize exterior noise (inner box, 54 × 35 × 35 cm; outer box, 68 × 50 × 51 cm). A condenser ultrasonic microphone (Ultramic250K; Dodotronic, Castel Gandolfo, Italy) was positioned at the center of the ceiling of the chamber. The USVs were recorded using the ultrasonic microphone with UltraSoundGate 416H data acquisition device (Avisoft Bioacoustics, Glienicke, Germany). The signals were band-filtered between 30 and 80 kHz for the 50 kHz USVs and analyzed using Avisoft-SASLab Pro (version 4.2, Avisoft Bioacoustics). Rats were habituated for 30 min in the chambers. After recording basal USVs for 30 min, the rats were given cocaine injection (15 mg/kg, i.p.) and/or acupuncture at HT7. The recordings were continued for a further 60 min. The number of USVs emitted at 10 min intervals was assessed.

### 4.7. In Vivo Fast-Scan Cyclic Voltammetry (FSCV) for DA Detection

Dopamine release in the NAc was measured by FSCV, as performed in our laboratory [29]. A custom-made carbon fiber electrode (CFE; 7 μm diameter, 150–200 μm length of exposed tip) was back-filled with 3 M KCl. The electrode potential was linearly scanned with a triangular waveform from −0.4~+1.3 V and back to−0.4 V versus Ag/AgCl using a scan rate of 400 V/s. Cyclic voltammograms were recorded at the CFE every 100 ms by means of a ChemClamp voltage clamp amplifier (Dagan Corporation, Minneapolis, MN, USA). Under urethane anesthesia (1.5 g/kg, i.p.), bipolar, stainless steel electrodes and CFE were stereotaxically positioned into the medial forebrain bundle (MFB; AP, −2.5 mm; ML, +1.9 mm; DV, 8.0 mm from the skull) and NAc core (AP, +1.6 mm; ML, +1.9 mm; DV, 8.0 mm from skull). MFB was stimulated with 60 monophasic pulses at 60 Hz at 2 min intervals. After a stable baseline was established (less than 10% variability in peak heights of 5 consecutive collections), the rats were given an intraperitoneal injection of cocaine (15 mg/kg) and/or acupuncture treatment at HT7. The changes of NAc DA release were recorded at 2 min intervals for 30 min after cocaine administration.

### 4.8. In Vivo Extracellular Recording of LH or Spinohypothalamic (SHT) Neurons

Extracellular single-unit recordings for LH or SHT neurons were performed, as described previously [30,31]. Briefly, urethane (1.5 g/kg, i.p.) was used to anesthetize the rats. For recording of LH neurons, a carbon-filament glass microelectrode (0.4–12 MΩ, Carbostar-1, Kation Scientific, Minneapolis, MN, USA) was stereotaxically inserted into the LH (stereotaxic coordinates: AP −1.8~−4.5 mm; ML 0.8~2.6 mm; DV 8.0~9.6 mm from the skull). After recording stable baseline for 10 min, the rats (*n* = 10) received acupuncture treatment at HT7 for 20 s and were recorded for 2 min after acupuncture. For recording of SHT neurons, the electrode was stereotaxically inserted into the spinal cord of C6~T2 and the wide dynamic ranges (WDR) neurons were identified by applying stimulation of brush, pressure and pinch. To verify SHT neurons, a stainless steel electrode was placed into the LH and antidromic stimulation was produced by passing a current (0.5 mA, 200 μs). The neuronal discharges from SHT neurons (*n* = 10 from 10 rats) were recorded during 20 s-HT7 stimulation. Single-unit activities were amplified and filtered at 0.1–10 kHz (ISO-80; World Precision Instruments, Sarasota, FL, USA), spikes were discriminated, and then event times were binned at 1 s intervals. The single-unit activities were recorded, and analyzed, using a CED 1401 Micro3 device and Spike2 software (Cambridge Electronic Design, Cambridge, UK).

### 4.9. Activation of the LH by PEPA Microinjection

PEPA (an AMPA receptor agonist; 3 ng/side) was injected to activate the LH as described previously [31]. Briefly, under pentobarbital anesthesia (50 mg/kg, i.p.) a double-barreled guide cannula (26 gauge, pedestal size 10 mm, Plastics One, Roanoke, VA, USA) was implanted bilaterally into the LH. The rats were allowed to recover for at least 7 days after operation. An internal cannula (33 gauge, Plastics One) that extended 1 mm beyond the tip of the guide cannula was min before being removed. Immediately after the injection, the rats received cocaine used to inject PEPA into the LH. PEPA was initially dissolved in 50% dimethyl sulfoxide (DMSO)/50% aCSF and diluted to 10% DMSO/90% aCSF immediately prior to use. PEPA was infused over 1 min (0.3 μL/min) and then the internal cannula was left in the guide cannula for 2 injection, and locomotor behaviors were measured up to 60 min after injection.

### 4.10. Immunohistochemistry for c-Fos

In a separate set of animals, rats (*n* = 13) were randomly assigned into Normal (*n* = 5), Cocaine (cocaine injection only, *n* = 4) and Cocaine + PEPA (PEPA injection into LH in cocaine treated rats, *n* = 4) groups. Sixty minutes after cocaine and/or PEPA injection, brains were taken out after perfusion with 4% paraformaldehyde, postfixed, and cryosectioned at 30 μm thickness. The sections were incubated with anti-c-Fos rabbit polyclonal antibodies (1:500; Sigma, USA), followed by incubation of a biotinylated donkey anti-rabbit Alexa Fluor 594 (1:500; Sigma). The slices were mounted on gelatin-coated slides, photographed, and quantified using a confocal laser scanning microscope (LSM700, Carl Zeiss, Jena, Germany). The number of c-Fos labeled cells was counted in a blindly chosen 290 μm × 300 μm area of the LH. The 4–6 brain pieces per animal were analyzed and averaged. Data were expressed as number of positive cells per group.

### 4.11. Data Analysis

All data are presented as the mean ± standard error of the mean (SEM) and analyzed by one-or two-way repeated ANOVA followed by *post hoc* tests using the Tukey method or Student’s *t*-tests, where appropriate. Statistical significance was considered at *p* < 0.05.

## Figures and Tables

**Figure 1 ijms-22-05994-f001:**
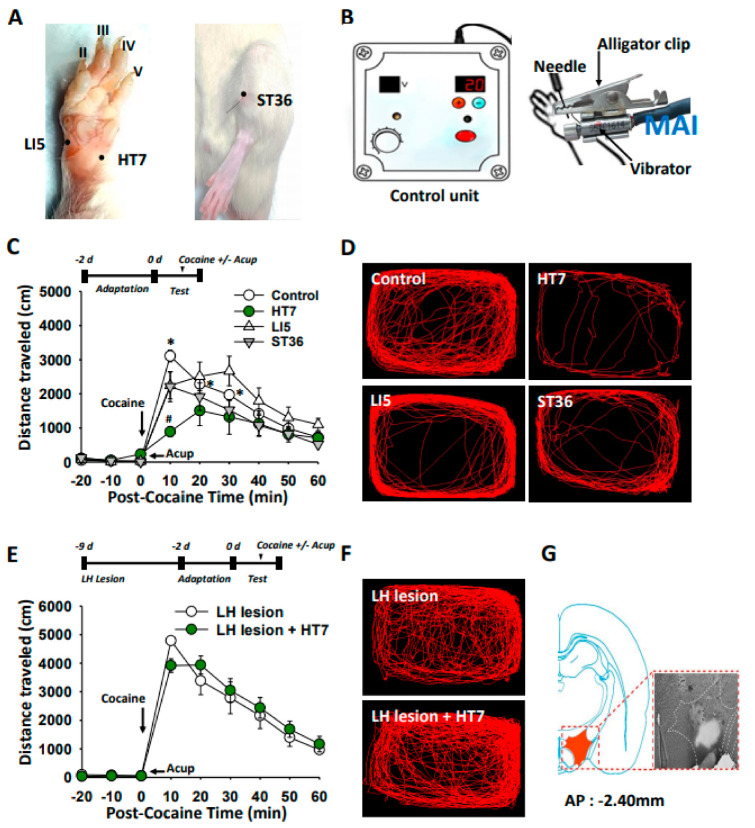
Effect of LH chemical lesions on acupuncture inhibition of cocaine-induced locomotor activity. (**A**) Schematic locations of the acupoints, HT7 and ST36, LI5. (**B**) Acupoints were stimulated with mechanical acupuncture instrument (MAI). (**C**,**D**) Effects of acupuncture on cocaine-induced enhancement of locomotor activity in rats. Rats received one acupuncture treatment for 20 s by using MAI. Acupuncture at HT7 (*n* = 6), but neither ST36 (*n* = 5) nor LI5 (*n* = 6), reduced the mean distance traveled per 10 min following cocaine injection, compared to Control group (cocaine injection only, *n* = 6; **C**). * *p* < 0.05 vs. the values before cocaine injection; ^#^
*p* < 0.05 vs. Control. Representative moving traces for 60 min after cocaine injection (**D**). (**E**–**G**) Effect of acupuncture on cocaine-induced locomotor activities in LH-lesioned rats. Representative moving traces for 60 min following cocaine injection in LH-lesioned rats (**F**). Toluidine blue-stained coronal sections of a rat brain with chemical lesions in the LH (**G**). LH lesion, cocaine injection in LH-lesioned rats (*n* = 6); LH lesion + HT7, acupuncture at HT7 after cocaine injection in LH-lesioned rats (*n* = 5). LH, lateral hypothalamus; Acup, acupuncture.

**Figure 2 ijms-22-05994-f002:**
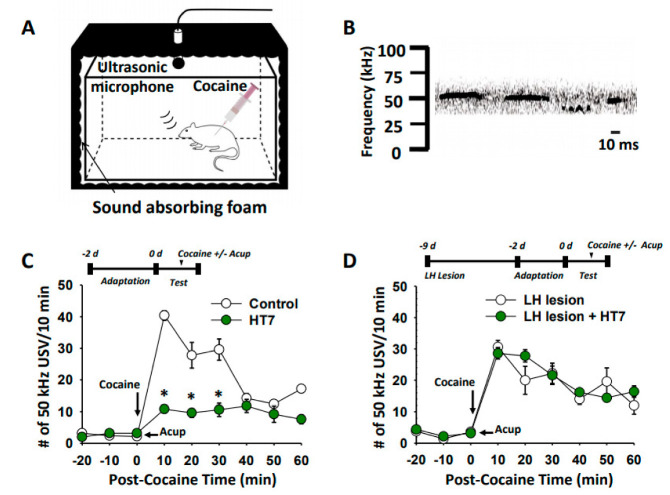
Effect of LH chemical lesions on acupuncture inhibition of cocaine-induced USVs. (**A**) Schematic diagram of ultrasonic vocalizations (USVs) recording in freely moving rats. (**B**) Representative traces of 50 kHz USVs following cocaine injection. (**C**) Effect of acupuncture at HT7 on cocaine-induced 50 kHz USVs in rats. Control, cocaine injection only (*n* = 5); HT7, acupuncture at HT7 in cocaine-treated rats (*n* = 5). * *p* < 0.05 vs. Control. (**D**) Effects of LH lesions on the effects of acupuncture at HT7 on cocaine-induced 50-kHz USVs**.** LH lesion, cocaine injection in LH-lesioned rats (*n* = 5); LH lesion + HT7, acupuncture at HT7 after cocaine injection in LH-lesioned rats (*n* = 5). Acup, acupuncture.

**Figure 3 ijms-22-05994-f003:**
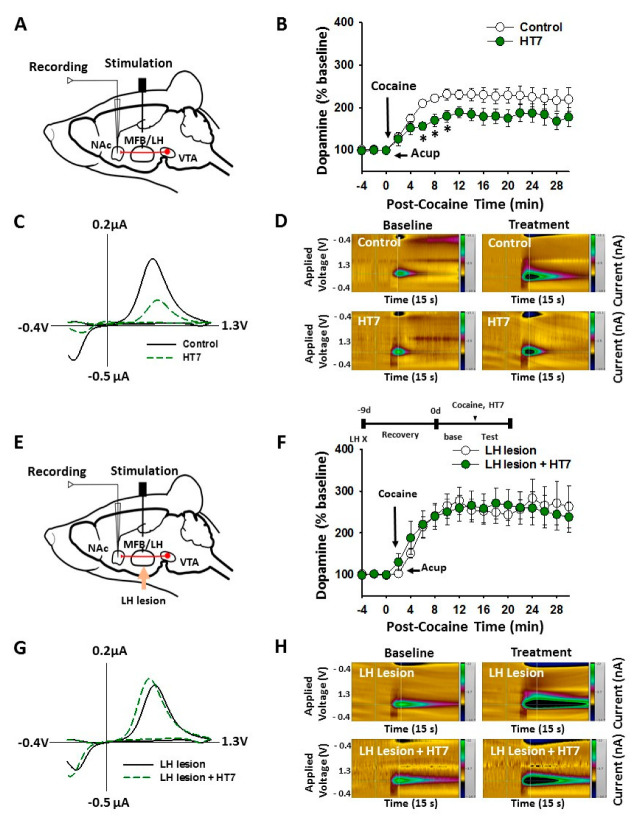
Mediation of the LH in acupuncture inhibition of cocaine-induced DA release in NAc. (**A**–**D**) Effect of acupuncture at HT7 on cocaine-induced DA release in NAc in rats. * *p* < 0.05 vs. Control. Schematic of fast-scan cyclic voltammetry (FSCV) for the recording of NAc DA release (**A**). (**B**) shows electrically stimulated NAc DA release at the 2 min intervals for 30 min after cocaine injection (15 mg/kg). * *p* < 0.05 vs. Control. Control, cocaine injection only (*n* = 9); HT7, acupuncture at HT7 in cocaine-treated rats (*n* = 7). Representative voltammograms (**C**) and pseudo-color plots (**D**) show DA FSCV signals 10 min after administration of cocaine. (**E**–**H**) Effect of LH lesions on the acupuncture inhibition of cocaine-induced DA release in NAc in rats. Schematic of FSCV for the recording of NAc DA release in LH-lesioned rats (**E**). (**F**) shows electrically stimulated NAc DA release at the 2 min intervals for 30 min after cocaine injection. Representative voltammograms (**G**) and pseudo-color plots (**H**) show DA FSCV signals 10 min after administration of cocaine. LH lesion, cocaine injection in LH-lesioned rats (*n* = 6); LH lesion + HT7, acupuncture at HT7 after cocaine injection in LH-lesioned rats (*n* = 5). Acup, acupuncture.

**Figure 4 ijms-22-05994-f004:**
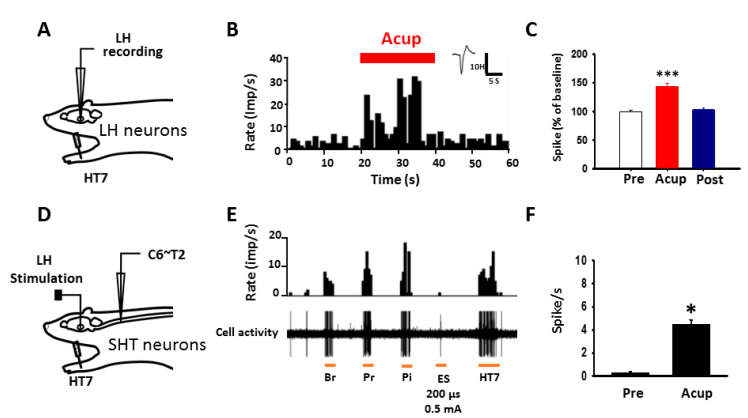
Excitation of LH or SHT neurons by acupuncture at HT7. (**A**) Schematic of in vivo extracellular recordings for LH neurons. (**B**,**C**) Effect of acupuncture at HT7 on single-unit activities of LH neurons. Peristimulus time histogram (**B**) and mean values of spikes before (Pre), during (Acup) and after (Post) acupuncture (expressed as percentage of pretreatment value, (**C**) *** *p* < 0.001 vs. Pre. *n* = 20. (**D**) Schematic of in vivo extracellular recordings for SHT neurons. (**E**,**F**) Effect of acupuncture at HT7 on single-unit activities of SHTN. Peristimulus time histogram (**E**) and mean values of spikes before (Pre) and during (Acup) acupuncture (expressed as percentage of pretreatment value, (**F**) * *p* < 0.05 vs. Pre. *n* = 10. Br, brush stimulation; Pr, pressure with a von Frey filament; Pi, pinch; Es, electric stimulation to LH; HT7, mechanical stimulation of needles inserted into HT7.

**Figure 5 ijms-22-05994-f005:**
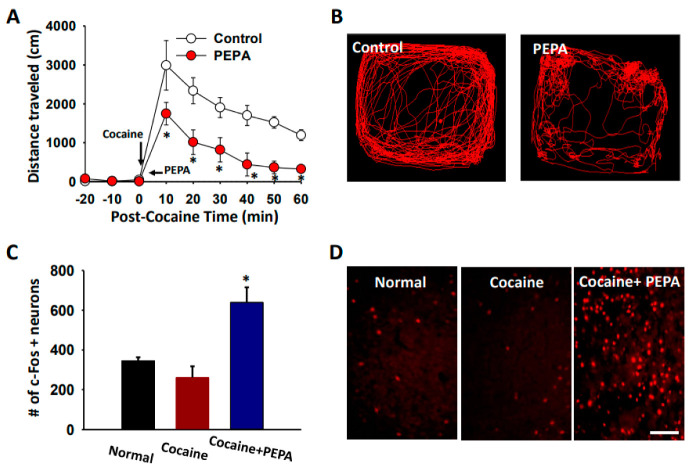
Reduction in cocaine locomotion by LH activation (**A**,**B**). Effect of PEPA injection on cocaine-induced locomotor activity in rats. PEPA microinjection into the LH reduced the mean distance traveled per 10 min following cocaine injection, compared to control group (cocaine injection only). Representative moving traces for 60 min after cocaine injection (**B**). Control, cocaine injection only (*n* = 5); PEPA, PEPA microinjection into the LH after cocaine injection (*n* = 5). * *p* < 0.05 vs. Control. (**C**,**D**) Enhanced c-Fos expression following PEPA injection into LH. Mean numbers of c-Fos positive neurons in control, cocaine and PEPA groups (**C**) and representative images of c-Fos expression in LH (**D**). Normal, normal rats (*n* = 5); Cocaine, cocaine-treated rats (*n* = 4); Cocaine + PEPA, microinjection of PEPA into LH in cocaine-treated rats (*n* = 4). * *p* < 0.05 vs. Normal or Cocaine. Bar = 100 μm.

## Data Availability

Not applicable.

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
