# Peer review of "Role of Lateral Hypothalamus in Acupuncture Inhibition of Cocaine Psychomotor Activity"

_ijms, 2021, doi:10.3390/ijms22115994_

Round 1

Reviewer 1 Report

This study demonstrates that acupuncture at HT7 inhibits cocaine-induced increases in behavioral and neurobiological readouts of psychosis by activating the spinohypothalamic (SH) pathway and ultimately, lateral hypothalamus (LH). This study is well-designed with multiple approaches to determine the involvement of LH activation by HT7 acupuncture in the cocaine-induced psychosis. I have some comments regarding the timing/duration of acupuncture and its long-lasting effect on the cocaine-induced psychosis.

  1. In Materials and Methods, it is not clearly described when the acupuncture was performed. According to figures, it appears that cocaine injection and the acupuncture were done almost at the same time. If the two manipulations were done together (or at a short interval), it raises a question regarding potentially differential effects of HT7 acupuncture on the initiation of psychosis and the maintenance. Although relatively fast, intraperitoneally given cocaine takes time to affect the brain, and thus, HT7 acupuncture may be affecting/suppressing the reward circuit even before the effect of cocaine is fully kicked-in. With respect to this, if HT7 acupuncture is given 20-30 min "after" cocaine administration, will it be still able to suppress cocaine-induced increases in, for example, locomotor behaviors and USV?
  2. The above question pertains to the effect of acupuncture on LH neuronal activity.  In Fig. 4, LH neurons are activated only when acupuncture is being given, not afterwards. How could this be reconciled with the long-lasting effect of the acupuncture on the cocaine-induced psychosis? Would this mean that the 20 sec activation of LH is sufficient to mitigate/prevent the effect of cocaine on the reward circuit? If so, does it depend on the timing of LH activation in relation to the cocaine administration?
  3. In Fig. 5, the locomotor traces for control and PEPA group are switched.
  4. Does the degree of LH neuronal activation by HT7 acupuncture is sufficient to induce c-Fos expression as PEPA?

Reviewer 2 Report

The manuscript showed the LH play a critical role in mediating ht7 acupuncture induced inhibition of caocaine-induced/addicted behavior in the rats. The approches are comprehensive and sound. The conclusion is clear. The findings are novel and would be significant in understaning the beneficial effect of HT7 acupuncture in treatment of cocaine addiction.  However, some information in Methods are missing, and the way of presentation in the figures can be further improved to avoid potential confusion. In Discussion, authors should probe the pysiological and clincial significance s of these finding. It would be better to state any limitations of this study. 

Comments

  1. Abstract: The 2nd sentence in Abstract seems to distract the stream of background ideas. Introduction
  2. Introduction:
  • the last sentence of the 1st paragraph: It would be better to introduce the two aspects, central vs. peripheral earlier. The idea of the last sentence seems to appear too abruptly.
  • Introduction: the last sentence of the 1st paragraph: (1) Authors may state the sentence starting with LH. ex) LH play any roles in....; (2) It would be better to add/state the fact that this idea is new.
  1. Materials and Methods
  • Cocaine treatment: need to provide the more information (see the comments in pdf file)
  • Line 81: need to specify the number of sessions and the time of the stimulation in relation to the time of cocaine administration and measurement of various parameters.
  1. Results
  • Line 196: The LH --à a circuit containing the LH
  • Figures 1~5: please find suggestions marked in the figure and its legend.
  • Line 213: need to provide the acupuncture at specific acupoint(s).
  • Line 219: please specify the surgery, bilateral or unilateral ?
  • Line 239: pleas find better word than ‘reduced’
  • Line 260: specify the number of WDR neurons.
  • Line 262: provide the baseline firing activity: xx+/- xx Hz (n=xx).
  • Line 275: the sentence need to be revised so a laymen can understand the meaning.
  • Line 277: revise the sentence.
  1. Discussion
  • Line 294; specify the acupuncture with acupoints
  • Line 300: The 2nd paragraph can be switched with the 3rd paragraph because it is mostly saying that the current work is consistent with previous findings. Whereas the 3rd paragraphs is about the novel finding of this study.
  • Line 321: The most salient finding of this study is revealing the involvement of LH area in HT7 acupuncture-induced inhibition cocaine locomotion, DA release, etc. Authors need to discuss whether this finding is consistent with previous reports and possible circuit model (Author may suggest a circuit diagram for HT acupuncture effect as a figure for discussion), physiological and clinical significance. Please also refer known and possible roles of the SHT-LH-NAc circuit in discussion.

Round 2

Reviewer 1 Report

The revision sufficiently addressed the reviewer's initial comments.